# Cognitive job crafting as mediator between behavioral job crafting and quality of care in residential homes for the elderly

**Marina Romeo[1], Montserrat Yepes-Baldó**  **[1]\*, Kristina Westerberg[2], Maria Nordin[2]**

**1** Research Group in Social, Environmental and Organizational Psychology (2014SGR992), Faculty of Psychology, Universitat de Barcelona, Barcelona, Spain, **2** Department of Psychology, Umeå Universitet, Umeå, Sweden

\* myepes@ub.edu

## Abstract

Extending previous studies on job crafting, the aim of the present study is to analyze the effect of job crafting on quality of care in residential homes for elderly people in two European countries (Spain and Sweden). We hypothesize that cognitive crafting could be a consequence of behavioral crafting and that it will mediate the relationship between behavioral crafting and the perception of quality of care. A correlational design was used, with two-waves approximately 12 months apart (n = 226). Our results indicate that behavioral job crafting at T1 had an effect on cognitive job crafting at T2, relational job crafting at T1 increases quality of care at T2, and the mediation effect of cognitive job crafting. These results indicate that we must differentiate between the two forms of crafting (behavioral and cognitive), not as indicators of the same latent construct, but as aggregates. Additionally, we point out two main implications for managerial practice. First, as relational job crafting has a direct effect on quality of care, it is important to assure an organizational culture oriented towards employees. Secondly, due to the mediation effect of cognitive job crafting, managers should facilitate meaningful work environments. To do so, jobs should be re-designed, increasing skills variety, identity and significance.

## Introduction

The recent European population projections indicate that the total population will increase from 511 million in 2016 to 520 million in 2070. Nevertheless, in the same period the working-age population (15–64 years old) will decrease significantly from 333 million in 2016 to 292 million [1]. The percentage of people over 65 years old will increase by 41.3%, and 12.5% of the total population will be represented by people over 80 years of age. In this light, the European Commission report draw two possible scenarios related to high life expectation; the positive one assumes that "all future gains in life expectancy are spent in good health" [1]. The other scenario, the negative one, associates the increasing years to health problems, and consequently with additional cost devoted to healthcare. In this scenario, the European Commission foresees that long-term care and health care costs are expected to contribute the most to the rise in age-related spending.

**Data Availability Statement:** All relevant data are within the manuscript and its Supporting Information files.

**Funding:** KW received funding from the Swedish Research Council for Health, Working Life and

Welfare - FORTE (Forte Dnr 2015-00708). https://forte.se/en/ The funders had no role in study design, data collection and analysis, decision to publish, or preparation of the manuscript.

**Competing interests:** The authors have declared that no competing interests exist.

Facing this situation, countries in Europe are concerned about how to make the welfare system sustainable, and additionally how to guarantee healthcare quality. In this sense, professionals are vital to the provision of health services and goods.

In Spain, the Catalan Association for Health Care Resources (Asociació Catalana de Recursos Asistencials, ACRA), which is formed by most enterprises and entities devoted to the assistance of elder people, indicated that this sector has more than 50,000 employees [2], and provide services for taxes equating 1,121.809,000 € in 2014 year's currency rate. In Sweden, according to the Council for the Promotion of Municipal Analysis [3], there is about 250,000 employees working in municipal eldercare and the corresponding total costs are 126,318.315,000 SEK in 2018.

Due to the high impact of this sector and its employees, from an economic and social point of view, it is important to point out that the last few years, the healthcare sector has been characterized by slow growth of costs, but healthcare professionals need an adequate amount of resources to reach their work-related goals and offer high-quality of care [4]. In this context, several authors [5] recognize the need for healthcare professionals to take a proactive role in shaping their future jobs to improve healthcare systems, to assure improvements on the quality of care they provide [6], and on other hand, this proactive role "*can be learned and effectively transferred from training to organizational practice*"[7 p321].

In this sense, the scientific literature has pointed out that the work environment can be proactively modified by the employees, who are not passive subjects, but can influence their workplace by redesigning their work experience from various processes. These processes have been called job crafting [8, 9].

The present study aims to analyze the effect of job crafting on quality of care in residential homes for the elderly people.

## Theoretical framework and hypotheses

Job crafting has been defined by Wrzesniewski and Dutton [9] as the "*physical and cognitive changes individuals make in the task or relational boundaries of their work*" (p179). Following Berg, Grant and Johnson [10] this proposal is the understanding that employees are often interested in customizing their jobs to fit optimally their motivations, competences and desires.

Organization's management should not unilaterally decide how its employees spend their time and energy. Rather, the employees themselves should be allowed to decide what to do, creating and searching a comfortable and enjoyable context, over and beyond the job descriptions provided by the management, particularly in complex and uncertain situations [9–12]. In this sense it is possible to consider that the job crafting emerges as a strategy for the leverage of work meaning and identity.

From the perspective of Wrzesniewski and Dutton [9], the concept of job crafting is based on three dimensions: a) the scope or amount of tasks that the employee develops (for example, the introduction of new tasks more suited to their abilities or interests; task job crafting), b) the relationships with other people and the activities that it carries out with the co-workers (for example, making friends with people with similar abilities or interests, relational job crafting), and c) the modification of the meaning of the labor and social environment (for example, recognize the effect and importance of work activity for users, patients, clients, etc., cognitive job crafting).

More recently, authors such as Zhang and Parker [13] grouped the three dimensions into two, called behavioral crafting (task and relational crafting) and cognitive crafting. The differentiation between behavioral and cognitive crafting arises from the debate between what

should be considered job crafting or not. In this sense, from the perspective of job demands and resources theory (JD-R), it is considered that cognitive crafting is a mere passive adaptation to work, so it cannot be conceived as crafting since it is not a proactive behavior of change [8, 14]. In contrast, Zhang and Parker [13], based on Wrzesniewski and Dutton [9] perspective, include cognitive job crafting since "*it involves altering how one frames or views their tasks or job, which is self-initiated, self-targeted, intentional, and represents meaningful changes to the job aspects*" [13 p5]. In this sense, it is important to differentiate between the two forms of crafting (behavioral and cognitive), not as indicators of the same latent construct, but as aggregates.

Preliminary research has indicated that cognitive crafting has only moderate relationships with behavioral crafting [15–17]. These results also reveal different relationship dynamics for task and relational job crafting, on the one hand, and cognitive job crafting, on the other. Additionally, recent studies also suggest that behavioral crafting (task and relational) can cause changes at the cognitive level [13,18]. Research within the field of social psychology demonstrates that individuals actively shape their reactions through cognitive reframing. Reframing is defined by Ashforth and Kreiner [19] as the transformation of the meaning assigned to an occupation by instilling it with positive value and neutralizing the negative values previously assigned.

In this sense, Unsworth, Mason, and Jones [18] and Zhang and Parker [13], based on the studies of Ashforth and Kreiner [19], point out that employees could change their perceptions about their tasks and relations at work (behavioral crafting) involving cognitive reframing (cognitive crafting). The authors [18] explained that employees reframe their cognitions in order to compare themselves with others in terms of their strengths. For example, if a group has productivity as a strength their members will tend to assess other groups or employees on the basis on this characteristic. In our research, following this suggestion, we consider that employees could cognitive reframe the meaning of their jobs by means of the tasks (Task crafting) they develop and the relationships they have at work (Relational crafting).

Related to the effects of job crafting, several empirical studies have demonstrated its influence on individual and organizational variables. On the individual level, job crafting has been positively related with employees' wellbeing [10, 11, 20, 21], engagement [22, 23], commitment [11] and job satisfaction [9]. On the other hand, on the organizational level, job crafting facilitates organizational change [24], improves performance [6, 25–28] and reduces turnover [11, 12, 21, 28].

In the healthcare sector, employees' job crafting contributes to employees' performance, analyzed as quality of care they give to patients [29]. During the past decades, performance indicators have become increasingly sophisticated [30], and the quality of care has been included as an indicator of performance [31]. Nevertheless, some authors [32] consider that it is difficult to compare quality of care and to transfer performance indicators directly between different health systems and cultures, because each country has established different indicators. However, the quality of care is used as an indicator of patients' outcome in studies focused in hospital care [33] and in the geriatric care [34].

Considering the complexity of the healthcare sector Teoh, Hassard, and Cox [35] developed a systematic review on the relationship between the working conditions and the quality of patient care. Results showed several investigations where an improved work environment exerted a positive effect on nurses assessed quality of care. For instance, promotion prospects, perceived salary and job security [36–38] were positively related with quality of care. Nevertheless, these authors concluded that the relationships between quality of care and work environment reported divergent findings, showing the complexity of these relationships [39, 40]. They recommended, among other aspects, to use multilevel or longitudinal designs, as well as

mediating and moderating variables, in order to present a more realistic interpretation of these relationships.

Following these suggestions and taking into account that job crafting is a modification of working conditions initiated by the employees, the present research propose job crafting as an antecedent of quality of care and a longitudinal design. Secondly, we propose cognitive crafting as a mediator between behavioral crafting (tasks and relationships), which modify working conditions, and quality of care.

Quality of care could be analyzed from the nurses' perception as an indicator of patient outcome [29, 33]. The present study uses this perspective to analyze this variable and, additionally, even though it is not unusual to assess quality of care with a single-item [41], we follow Westerberg and Tafvelin [31] recommendations, adding information about satisfaction "*with the way in which the clients were treated, kept informed and their wishes respected*" [p464], and use a 5-items scale.

The results of previous studies [9–11] reveal different relationship dynamics for task and relational job crafting, on the one hand, and cognitive job crafting, on the other. In light of the revisions of Unsworth et al. [18] and Zhang and Parker [13], there exist several empirical and theoretical reasons to hypothesize that cognitive crafting could be a consequence of behavioral crafting and that it will mediate the relationship between behavioral crafting and the perception of quality of care. Compared to previous studies, the main novelty of our proposal lies in the fact that we analyze job crafting as a dynamic process and differentiate between behavioral job crafting (tasks and relational crafting), as an antecedent, and cognitive job crafting as a consequence, and as a mediator between it and the quality of care.

Consequently, the following hypothesis will be tested:

H1. Behavioral Job Crafting (JC) (T1) is positively related with Cognitive JC (T2)

H1.1. Task JC (T1) is positively related with Cognitive JC (T2)

H1.2. Relational JC (T1) is positively related with Cognitive JC (T2)

H2. Behavioral JC (T1) is positively related with Quality of Care (T2)

H2.1. Task JC (T1) is positively related with Quality of Care (T2)

H2.2. Relational JC (T1) is positively related with Quality of Care (T2)

H3. Cognitive JC (T2) mediates the relationship between Behavioral JC (T1) and Quality of Care (T2)

H3.1. Cognitive JC (T2) mediates the relationship between Task JC (T1) and Quality of Care (T2)

H3.2. Cognitive JC (T2) mediates the relationship between Relational JC (T1) and Quality of Care (T2)

Fig 1 illustrates the variables and hypotheses that constitute our research model.

## Materials and methods

### Participants and procedure

Data was collected in 18 residential homes for elderly people in two European countries (Spain and Sweden). A correlational design was used, with two-waves approximately 12 months apart.

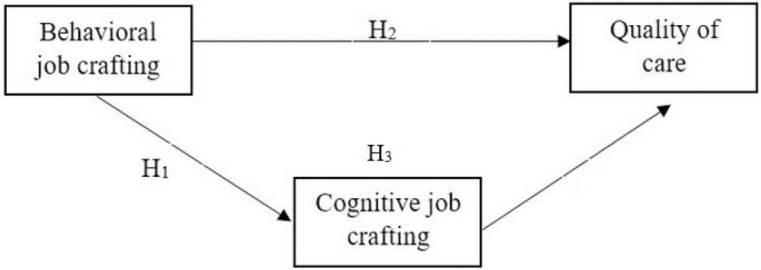

**Fig 1. Research model.**

The protocol for the research project has been approved by the managers and research ethics committees of the participating centers, in Spain (due to confidentiality agreements, the complete names of the participating centers cannot be disclosed), and the regional ethics committee in Sweden (Regional Board of Ethics, EPN, Umeå [ref 2015-62-31Ö]). All participants gave informed consent for the research, and that their anonymity was preserved.

At time 1 (2016), participants received information about the study aim and ethical considerations, and the written questionnaire to be completed. Those who returned usable surveys were contacted again at time 2 (2017).

At T1 and T2, participants provided data about their age, gender, if they have a permanent contract, a managerial position, and worked night shift. The questionnaire included self-reported measures of behavioral job crafting (task and relational job crafting), cognitive job crafting and perceived quality of care. Participants were asked to add a secret code that allowed us to match responses from T1 and T2 without identifying individual employees. The application format was "paper and pencil" in all centers.

Of the 928 contacted employees at T1, 628 returned complete questionnaires (68% response rate), and at T2 890 employees were contacted and 501 answered (56% response rate). A total of 226 employees answered at both T1 and T2. Employees in the final sample (n = 226) were predominantly female (92.5%), with a mean age of 45.76 years (SD = 11.42), 92.5% of them had permanent contracts, 10.2% had managerial positions and 10.2% worked on shifts with night shifts. Most respondents worked in Spain (65.93%)

## Materials

Two instruments were used in this study to evaluate the variables that constitute our research model. To measure job crafting, the Job Crafting Questionnaire (JCQ) [16] was used, both for behavioral and cognitive crafting. Recently, a new Spanish adaptation of this questionnaire has been developed [42]. The Spanish versions of the instrument used in the present study [6] and in the newest one [42] are very similar. Only small changes on translation have been observed. The internal consistency for the Spanish version of this scale was .85 [6] (.88 in the newest version [42]), and .83 for the Swedish version [6]).

On the other hand, for measuring quality of care the instrument developed by Westerberg and Tafvelin [31] was used.

*Behavioral job crafting* (10 items). Task (TJC) and Relational (RJC) Job Crafting dimensions of the Job Crafting Questionnaire (JCQ) [16] were used to assess Behavioral Job Crafting (BJC). Responses were measured on a five-point Likert scale ranging from 1, "hardly ever", to 5, "very often". An example of item for TJC is "How often do you change the scope or types of tasks that you complete at work?" and for RJC "How often do you make friends with people at

work who have similar skills or interests?" Task (TJC) Job Crafting subscale had a Cronbach's alpha of .87 and Relational (RJC) Job Crafting .83 [16].

*Cognitive job crafting* (5 items). Cognitive Job Crafting dimension of the Job Crafting Questionnaire (JCQ) [16] was used to assess Cognitive Job Crafting (CJC). Responses were measured on a five-point Likert scale ranging from 1, "hardly ever", to 5, "very often". An example of item is "How often do you remind yourself of the importance of your work for the broader community?" Cognitive (CJC) Job Crafting subscale had a Cronbach's alpha of .89 [16].

*Quality of care (QoC)*. The instrument developed by Westerberg and Tafvelin [31] was used. It includes three statements about how often the respondent felt satisfied with the way in which the clients were treated, kept informed, and their wishes respected, one statement about how well the help and support provided met the client's needs, and another item related to the overall satisfaction with care work. An example of item is "At my workplace I experience that enough consideration is taken to the users'/clients' opinions and wishes". Responses were measured on a five-point Likert scale ranging from 1, "very seldom or never", to 5, "very often or always". The internal consistency of the scale was .86 [31].

*Control variables*. We controlled for the possible effects of participants' age, gender, contract, managerial position, and shift.

## Results

### Descriptive statistics

Descriptive statistics for all the study variables in both waves are included in Table 1. No significant differences were found between T1 and T2 variables' scores, indicating the stability of the measures. In all cases, the scores for job crafting and quality of care variables were above 3 in a 5-points scale. Following the scale range these results indicated that employees sometimes job craft, and often considered that quality of care in their workplaces was appropriate.

Regarding correlations among the variables, task job crafting (T2) was not associated with quality of care, neither at T1 nor at T2. All other variables had significant relationships among them. Additionally, all scales showed good reliability values, above .70 (Table 2).

### Test of hypotheses

To test H1 and H2 hierarchical linear regression analyses were performed. As a first step, control variables were included. Secondly, task (H1.1) and relational (H1.2) job crafting in T1 were introduced as antecedents of cognitive job crafting in T2. Similarly, task (H2.1) and relational (H2.2) job crafting in T1 were introduced as antecedents of quality of care in T2.

**Table 1. Descriptive statistics at time 1 (T1) and time 2 (T2).**

|  | N | Minimum | Maximum | Mean | SD | Paired t-test |
|---|---|---|---|---|---|---|
| QoC T1 | 226 | 1.80 | 5.00 | 4.04 | .60 | .131 (ns) |
| QoC T2 | 225 | 2.00 | 5.00 | 4.03 | .61 | |
| TJC T1 | 224 | 1.00 | 5.00 | 3.22 | .75 | .633 (ns) |
| TJC T2 | 223 | 1.00 | 5.00 | 3.20 | .73 | |
| RJC T1 | 226 | 1.00 | 5.00 | 3.36 | .80 | .646 (ns) |
| RJC T2 | 224 | 1.00 | 5.00 | 3.32 | .82 | |
| CJC T1 | 226 | 1.00 | 5.00 | 3.19 | .82 | -.948 (ns) |
| CJC T2 | 223 | 1.00 | 5.00 | 3.23 | .80 | |

Note: QoC, Quality of care; TJC, Task Job Crafting; RJC, Relational Job Crafting; CJC, Cognitive Job Crafting; SD, Standard Deviation; ns, no significant relationship.

**Table 2. Correlations and alpha coefficients.**

| Variables | 1 | 2 | 3 | 4 | 5 | 6 | 7 | 8 |
|---|---|---|---|---|---|---|---|---|
| 1. QoC T1 | .811 | | | | | | | |
| 2. QoC T2 | .575** | .838 | | | | | | |
| 3. TJC T1 | .181** | .174** | .777 | | | | | |
| 4. TJC T2 | .056 | .115 | .519** | .753 | | | | |
| 5. RJC T1 | .353** | .240** | .386** | .224** | .761 | | | |
| 6. RJC T2 | .265** | .319** | .274** | .359** | .595** | .717 | | |
| 7. CJC T1 | .267** | .155* | .465** | .358** | .438** | .338** | .842 | |
| 8. CJC T2 | .265** | .226** | .378** | .421** | .330** | .445** | .700** | .866 |

$**p < 0.01$. Scale alpha coefficient in the diagonal.

Note: T1, Time 1; T2, Time 2; QoC, Quality of care; TJC, Task Job Crafting; RJC, Relational Job Crafting; CJC, Cognitive Job Crafting.

To test H3, PROCESS macro [43] was used, as it generates total, direct and indirect effects in simple mediation models. We estimated model 4 in PROCESS, with 10,000 bootstrap samples and 95% bias-corrected bootstrap confidence intervals for all indirect effects, controlling for all control variables. We conducted separate analyses for task job crafting (T1) and relational job crafting (T1). Cognitive job crafting in T2 was used as mediator on the relationship between behavioral crafting in T1 and quality of care in T2.

Our results supported H1, as task ($\beta_{TJC1} = .373$, $p < .001$; Corrected $R^2 = .138$. F = 4.069, p = .001) and relational job crafting ($\beta_{RJC1} = .406$, $p < .001$; Corrected $R^2 = .16$. F = 4.718, $p < .001$) in T1 were positively related with cognitive job crafting in T2. No significant effect of the control variables (gender, age, permanent contract, managerial position, night shift) was found.

H2 was partially supported, as only relational job crafting at T1 influenced quality of care at T2 (H2.2) ($B_{RJC1} = .286$, $p = .002$; Corrected $R^2 = .071$. F = 2.501, p = .026). H2.1 was not supported ($\beta_{TJC1} = .166$, ns; Corrected $R^2 = .021$. F = 1.405, p = .219), and in both cases no significant effect of the control variables (gender, age, permanent contract, managerial position, night shift) was found.

Finally, both mediation models were supported (H3). We observed a significant indirect effect of task job crafting at T1 (indirect effect = .0601, $p < 0.01$; SE = .0259; 95% CI: .015 to .1156) (Fig 2) and relational job crafting in T1 (indirect effect = .0404, $p < 0.001$; SE = .0203; 95% CI: .0056 to .0866) (Fig 3) on quality of care at T2 through cognitive job crafting at T2.

## Discussion

### Implications for theory and research

The purpose of the present study was to analyze the effect of job crafting on quality of care in residential homes for elderly people in two European countries (Spain and Sweden). Our

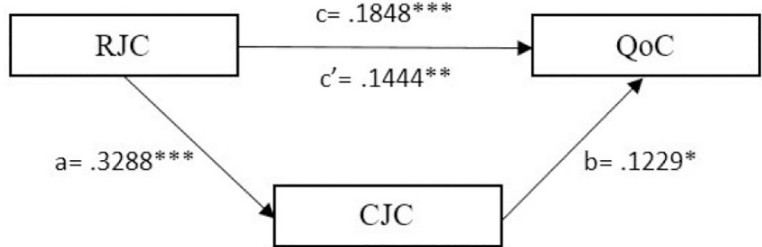

**Fig 2. Simple mediation model of CJC (T2) between RJC (T1) and QoC (T2).** Scores are not standardized.
$*p < 0.05$; $**p < 0.01$; $***p < 0.001$.

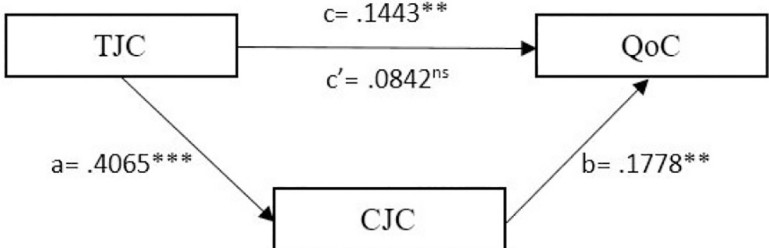

**Fig 3. Simple mediation model of CJC (T2) between TJC (T1) and QoC (T2).** Scores are not standardized.
$^{**}p < 0.01$; $^{***}p < 0.001$; ns: non-significant.

research extends previous studies on job crafting from the perspective of Wrzesniewski and Dutton [9], considering task and relational crafting as behavioral crafting [13], and as antecedents of cognitive crafting.

Additionally, and contrary to the perspective of job demands and resources theory (JD-R) [8, 14], the present research on cognitive job crafting is not considered as a mere passive adaptation to work, but as a proactive behavior of change, as pointed out by Wrzesniewski and Dutton [9] and Slemp and Vella-Brodick [16]. It is considered that employees could change their perceptions about their tasks and relations at work (behavioral crafting) by involving cognitive reframing (cognitive crafting) [13, 18]. Consequently, this process could affect performance [9, 12, 29, 33, 34].

As an added value of this research, we take measures in two waves, to account for the dynamic nature of the phenomenon. Some studies have previously used longitudinal designs but the majority of them used the JD-R model of job crafting [14], as it is the most common model for studying job crafting [6, 44], and none of them introduced cognitive job crafting as consequence of behavioral job crafting. Our results supported the hypothesis that behavioral job crafting at T1 had an effect on cognitive job crafting in T2, as demonstrated by correlations and regression analyses performed.

This result is in line with other studies [15–17], which used general employed population samples and cross-sectional designs. However, a second longitudinal study by Niessen et al. [15], found no relationship between behavioral job crafting at T1 and cognitive job crafting at T2. The authors indicated that the time lag of two weeks may be too short, while in the present study we lasted a year between measures, following the recommendations of other researchers [45].

Secondly, our results supported partially the hypothesis that behavioral job crafting increases quality of care. Specifically, correlations between task job crafting and relational job crafting in T1 with quality of care in T2 were positive and significant, having relational job crafting in T1 the highest relation with quality of care in T2. Nevertheless, regression analyses have shown that only relational job crafting at T1 had a significant effect on quality of care at T2.

Research has shown that individuals who have a strong need for relatedness tend to have collectivist tendencies [46] and to help group members [47]. Leana et al. [12] consider that job crafting can also be a collaborative activity, as it "*involves joint effort among employees in the service of changing work process*" (p1173). It "*is not the work of an individual agent, as described by Wrzesniewski and Dutton, but instead is the work of a dyad or group of employees who together make physical and cognitive changes in the task or relational boundaries of their work*" [12 p1173].

Finally, we tested the mediation effect of cognitive job crafting at T2 on the relationship between behavioral job crafting in T1 and quality of care at T2. We observe different results depending on the independent variable (task or relational crafting). In the case of relational job crafting, the total and the direct effects are positive and significant, while in the case of task job crafting only the total effect is significant. Nevertheless, in both cases the results confirmed our hypotheses related to mediation effect of cognitive job crafting (indirect effect).

This result is in relation with our correlational and regression analyses, where only relational job crafting had a significant effect on quality of care. Previous cross-sectional studies [29] indicated that task job crafting had the lowest correlation coefficients with quality of care, even though they were statistically significant. The same study indicated that task and relational job crafting explained similar amounts of variance of quality of care (around 3.9%).

The current research has both, theoretical and practical implications, for understanding the psychological mechanisms underlying the association between behavioral job crafting and quality of care on residential homes for elderly people. To be more specific, the present study develops the perspective of Wrzesniewski and Dutton [9] on job crafting by conceptualizing it as a dynamic process where the three components of the model have different roles when explaining quality of care perceptions.

In this sense, our results suggest that we must differentiate between the two forms of crafting (behavioral and cognitive), not as indicators of the same latent construct, but as aggregates [48], unlike the original model from Wrzesniewski and Dutton [9] and Slemp and Vella-Brodrick [16], who conceive them as indicators of the same supraordinal construct.

In relation to Tim's and Bakker's model [28], our study provides evidence on the role that cognitive job crafting plays. Cognitive job crafting is not included in the mentioned model but, according to our study, it mediates the relationship between behavioral crafting, which could be equated with increasing social and structural job resources, and even with the increasing challenging job demands [13], and quality of care.

Our research assesses the effects of a positive organizational intervention. Specifically, our findings indicate that organizations may foster quality of care facilitating job crafting. In this sense, a practical implication of our research is that promoting job crafting in the healthcare sector might be worthwhile. The possibility of allowing workers to job craft should be highly considered by managers and organizations within the healthcare sector, especially if economic contractions are experienced, and considering that the healthcare professionals are working in a demanding environment. Following a psychosocial perspective, we considered important to point out the context key aspects for human resource management. In this sense, it is important to guarantee the working conditions that allow employees turn the job they have into the job they want [49].

Additionally, it is possible to point out the implications for job design, because "*recognizing that the strength of individual needs varies across employees and allowing them the opportunity to adjust tasks, relationships, and skills in ways that enable need-fulfillment at work is important*" [50 p624].

Managers must acknowledge and understand employees' perspectives, encourage self-initiative and minimize control wherever possible. In this sense, our research adds empirical evidence at the self-determination theory [51], suggesting how job crafting can facilitate self-satisfaction of employees' needs. Managers should be aware about their employees' needs and try to encompass employees' needs into those behaviors that are most desirable for the organization [52]. In order to achieve this objective, it is important to assure an organizational culture oriented towards employees [53, 54], as relational job crafting has a direct effect on quality of care. In this sense, as pointed out by Berdicchia, Nicolli, and Masino [55] from the JD-R perspective, "*managerial interventions aimed at changing the organizational culture may*

*be useful to lower perceived social costs and increase the perceived instrumental value of social exchanges*" (p 327).

Secondly, due to the mediation effect of cognitive job crafting, managers should facilitate meaningful work environments. The implications related this result for job design are clear, because to do so, jobs should be re-designed, increasing skills variety, identity and significance [56, 57].

## Limitations and future research

Despite the interesting findings we have obtained, our study has some limitations to take into account. First, this research has limitations related to the psychosocial and individual factors considered. Several authors have pointed out as antecedents of job crafting proactive personality [8, 58–62], knowledge, skills, and abilities [62], self-efficacy [58], social support [58], and situational features of accountability, ambiguity, and autonomy [59]. Future research should include some of these variables, either as possible predictors or as control variables.

Secondly, sample characteristics (size and organizational context). This limits the external validity of our findings, as we collected data from twelve organizations from two European countries, even though they represent the north and south of Europe with their contextual particularities. Although the sample obtained cannot be considered as representative due to the non-probability selection system, the demographic data for this sample is similar to that available in Spain (ACRA) [2] and Sweden [3] in terms of gender (mostly women), age, and type of work. In future research other countries should be included, as well as measures about the possible effects of organizational context and/or culture. Additionally, separate analyses by country should be done, but the small Swedish sample (n = 77; 34.1%) in the present research did not allow it.

Thirdly, we collected data on two waves. It would be interesting to have a third wave to confirm the mediation results strongly and to test new hypothesis. In this sense, as a dynamic process, we understand that there can be a reciprocal effect between variables, so the behavioral job crafting would affect the cognitive job crafting and vice versa. Having a third wave would allow to test this hypothesis. Nevertheless, as we have explained previously, the response rate in T2 was 33.3%, decreasing in almost 45%, and the participant organizations declined to collect a third wave.

Fourthly, we focus on the perception of employees' job crafting and quality of care. Related to job crafting, future research must consider the quality and sustainability of the job crafting developed by employees, in order to tackle the phenomenon in greater depth. Related quality of care, although it is acceptable to collect employees' perception as an indicator of patient outcome [33, 34] using self-reports, some authors recommend to include alternative external assessment as well as measures about relational and functional aspects of quality [63], providing greater levels of objectivity. In this sense, future research should include employees, managers, users and relatives' perceptions of quality of care, as well as objective measures as survival (mortality), incidence of disease (morbidity), health-related quality of life issues (ulcer rates, number of users falls), or ratio of residents per employee.

Finally, a limitation of PROCESS is that only one exogenous variable can be entered in a single analysis. That is the reason why we used 2 separate mediation models. Additionally, as we aforementioned, previous research has arisen that the 3 components of job crafting have different relationship dynamics for task and relational job crafting, on the one hand, and cognitive job crafting, on the other. Additionally, as pointed out by Hayes et al. [64], SEM is more suitable with large samples, as it relays on large sample asymptotic theory. For this reason, as we have a sample of 226 participants, we decided to use PROCESS. Nevertheless, in future

research with larger samples, the data would be more parsimoniously handled with SEM models.

## Supporting information

**S1 File. Data set.**
(SAV)

## Author Contributions

**Conceptualization:** Marina Romeo, Montserrat Yepes-Baldó, Kristina Westerberg, Maria Nordin.

**Data curation:** Montserrat Yepes-Baldó, Kristina Westerberg.

**Formal analysis:** Marina Romeo, Montserrat Yepes-Baldó, Kristina Westerberg, Maria Nordin.

**Funding acquisition:** Kristina Westerberg.

**Investigation:** Marina Romeo, Montserrat Yepes-Baldó, Kristina Westerberg, Maria Nordin.

**Methodology:** Marina Romeo, Montserrat Yepes-Baldó, Maria Nordin.

**Project administration:** Marina Romeo.

**Supervision:** Marina Romeo, Kristina Westerberg.

**Writing – original draft:** Marina Romeo, Montserrat Yepes-Baldó, Kristina Westerberg, Maria Nordin.

**Writing – review & editing:** Marina Romeo, Montserrat Yepes-Baldó, Kristina Westerberg, Maria Nordin.

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
