## [Decision Letter · Decision Letter 0]

1 Apr 2020

PONE-D-20-01554

Cognitive job crafting as mediator between behavioral job crafting and quality of care in residential homes for the elderly

PLOS ONE

Dear Dr. Yepes-Baldó,

Thank you for submitting your manuscript to PLOS ONE. After careful consideration, we feel that it has merit but does not fully meet PLOS ONE’s publication criteria as it currently stands. Therefore, we invite you to submit a revised version of the manuscript that addresses the points raised during the review process.

We would appreciate receiving your revised manuscript by May 16 2020 11:59PM. To enhance the reproducibility of your results, we recommend that if applicable you deposit your laboratory protocols in protocols.io, where a protocol can be assigned its own identifier (DOI) such that it can be cited independently in the future. For instructions see: http://journals.plos.org/plosone/s/submission-guidelines#loc-laboratory-protocols

We look forward to receiving your revised manuscript.

Kind regards,

Anthony Montgomery

Academic Editor

PLOS ONE

Journal Requirements:

Reviewers' comments:

Reviewer's Responses to Questions

**Comments to the Author**

1. Is the manuscript technically sound, and do the data support the conclusions?

Reviewer #1: Partly

Reviewer #2: Partly

2. Has the statistical analysis been performed appropriately and rigorously? 

Reviewer #1: Yes

Reviewer #2: No

3. Have the authors made all data underlying the findings in their manuscript fully available?

Reviewer #1: Yes

Reviewer #2: No

4. Is the manuscript presented in an intelligible fashion and written in standard English?

Reviewer #1: Yes

Reviewer #2: No

5. Review Comments to the Author

Reviewer #1: PONE-D-20-01554: Cognitive job crafting as mediator between behavioral job crafting and quality of care in residential homes for the elderly

The present study was a prospective study of job crafting in the residential care industry, examining whether cognitive crafting mediated the relation between behavioral crafting and quality of care. Authors concluded that cognitive crafting was a mediator between these other variables. A strength of the study was the use of prospective data and the unique sample. Notwithstanding these strengths, I have some concerns about the paper that I outline below in case the authors might use the feedback to improve their work:

1. The introduction was too brief and did not adequately build a case for why a job crafting study was needed in this population (residential care staff), and similarly, why cognitive crafting would act as a mediator between the more behavioral aspects of crafting and quality of care. These needs to be unpacked quite substantially, as far too much is left to the reader currently. Questions I had were:

a. Why is cognitive crafting the mediating variable? Why not another dimensions of crafting?

b. Could cognitive crafting precede the more behavioral aspects of crafting?

c. Why wouldn’t all three aspects of job crafting be correlated predictors that are theorized to occur simultaneously?

d. Why residential care staff? I realise they are an important population and the statistcs presented at the beginning of the manuscript illustrate that point, but my question was more around why job crafting would be a particularly helpful strategy with this population? Can you link the two in some way? I feel this would be instrumental in helping to build a more persuasive case for the contribution of the study

2. Regarding the method, please add the inclusion and exclusion critieria for the participants. Also, how do the characteristics of this sample match the population of residential care staff in either Spain or Sweden? To they approximate these groups?

3. There were some aspects of the measures that were not well described. For example, I was confused by this sentence: "The internal consistency for the different Spanish versions of this scale was .85 [32] and .88 [31], and .83 for the Swedish version [32]." Is that in the present study, or in the original publications? What about the validity of the measures?

4. For the analyses, it was odd that the models were tested as separated mediation models using PROCESS. I think the data would be more parsimoniously handled with SEM models, where you could a) include all variables simultaneously, and b) account for attenuation caused by measurement error when you model with latent variables, and c) test competing positions of the variables so that you could better establish which one is better position as a mediator, predictor, etc.

5. On that point above, I would like to see these models compared against a model in which all three job crafting dimensions were treated as the predictor of quality of care. If this mediation model was better, then it at least provides some empirical support behind the authors’ reasoning/hypotheses, but currently there is not much. I recommend SEM as the approach to do this.

6. There is a bit too much describing of the results in the discussion. I was more interested in how this all fits with the existing literature, and a deeper discussion of the practical and theoretical implications. Especially the practical limitations feel very light, with very few details provided. I suggest you expand both of these sections considerably.

All the best with your research!

Reviewer #2: Dear Authors,

Thank you for submitting your manuscript and for the opportunity to review it. I certainly think it has merit but there are a number of recommendations I believe you should consider in order to improve it.

1. The Introduction is somewhat descriptive and could be more critical in reviewing the existing literature. First and foremost, there needs to a clear explanation into what Job Crafting is and why it is important. In particular, the manuscript alludes to a difference between the JDR perspective of crafting and that of Wrzesniewski and Dutton which is used here – however unless a reader is familiar with the job crafting literature many will not know the difference. There isn’t a clear explanation on why job crafting is important.

2. Similarly, how does this fit in with existing theoretical models. Much of the Introduction describes existing relationships (particularly from Line 102 onwards on Page 5), but why do these relationships happen? What is the process behind it? This is particularly the case between crafting and performance (Tims et al., 2015). It would also be useful to reflect on how quality of care is different/ same to existing performance measures, and how this links in with the wider psychosocial working environment (see Teoh et al., 2019).

3. Could you please expand and clarify the sentence “It is considered that cognitive crafting is a mere passive adaptation to work, so it cannot be conceived as crafting since it is not a proactive behavior of change” please? I’m not familiar with this perspective. From what I understand it appears that Zhang and Parker are saying that it isn’t job crafting, but then in the next sentence argues that is still is?

4. Page 6, 113. What are the recommendations from Westerberg and Tafvelin and how is it relevant to the point being made? When I first read this paragraph I understood it that the intention was to use a single item measure of quality of care, but this is not the case as the study uses a 5 item measure instead.

5. Page 6, line 115. It would be useful to consider reiterating these reasons, or then making them clearer earlier on, as it is not really evident.

6. Its great, a real strength, to be able to carry out a longitudinal study. Nevertheless there needs to be a rationale and description of this in the Introduction. In particularly, within the hypotheses need to make clear whether the measures are at T1 or T2 because as they currently are this appears to be a cross-sectional study.

7. Within the material section, it is not clear if the surveys administered were done so in Swedish or in English. For the JCQ in Spanish, I don’t understand the two different Spanish versions. There is a Spanish adaption [31] that wasn’t used which I presume because it came out after the study began? Bu then the version that was used in this study, was also used in a previous study [4], so why is there a need to mention the new version develop in [31]?

8. It isn’t clear if the Cronbach Alpha’s in the material section refers to that from previous studies or from the current study. I understand it as the former but I am not sure.

9. Page 9, Line 197 – what does medium to high mean? Are there established thresholds these are compared against?

10. I believe that the data from Sweden and Spain were mixed-together which is fine, but I think it would be important to control for this within the analysis. Considering the longitudinal design, the relationships between cognitive crafting and quality of care still are cross-sectional. At the least, cognitive crafting at T1 should be tested as a predictor of quality of care at T2. Building on the Introduction, there needs to be some rationale as to why cognitive at crafting at T2 was used as the mediator (opposed to at T1).

11. Considering that job crafting is strongly influenced by psychosocial and individual factors (Rudolph et al., 2016), I am left to wonder if these could/should have been included within this study. Either as possible predictors or then as control variables. At the very least, they should be acknowledged within the manuscript and discussed.

I hope you find the comments above constructive and I wish you all the best.

References

Rudolph CW, Katz IM, Lavigne KN, Zacher H. Job crafting: A meta-analysis of relationships with individual differences, job characteristics, and work outcomes. J Vocat Behav. 2017;102(314):112–38.

Teoh, K., Hassard, J., & Cox, T. (2019). Doctors’ perceived working conditions and the quality of patient care: a systematic review. Work & Stress, 33(4), 385-413.

Tims M, Bakker AB, Derks D. Job crafting and job performance: A longitudinal study. Eur J Work Organ Psychol. 2015;24(6):914–28

6. PLOS authors have the option to publish the peer review history of their article (what does this mean?). If published, this will include your full peer review and any attached files.

Reviewer #1: No

Reviewer #2: No

---

## [Author Response · Author response to Decision Letter 0]

7 May 2020

As indicated in the Response to reviewers, we include here our answers:

Reviewer #1: PONE-D-20-01554: Cognitive job crafting as mediator between behavioral job crafting and quality of care in residential homes for the elderly

The present study was a prospective study of job crafting in the residential care industry, examining whether cognitive crafting mediated the relation between behavioral crafting and quality of care. Authors concluded that cognitive crafting was a mediator between these other variables. A strength of the study was the use of prospective data and the unique sample. 

Thank you for your comments!

Notwithstanding these strengths, I have some concerns about the paper that I outline below in case the authors might use the feedback to improve their work:

1. The introduction was too brief and did not adequately build a case for why a job crafting study was needed in this population (residential care staff), and similarly, why cognitive crafting would act as a mediator between the more behavioral aspects of crafting and quality of care. These needs to be unpacked quite substantially, as far too much is left to the reader currently. Questions I had were:

a. Why is cognitive crafting the mediating variable? Why not another dimensions of crafting? 

Zhang and Parker (2018) analyze Bruning and Campion's (2018) proposal, showing that behavioral and cognitive crafting are closely related and are not mutually exclusive dimensions. 

The differentiation between behavioral and cognitive crafting arises from the existing debate between what should or should not be considered job crafting. Precisely, from the perspective of demands and resources, cognitive crafting is considered a mere passive adaptation to work, so it cannot be conceived as crafting since it is not a proactive behavior of change (Bakker, Tims, & Derks, 2012; Tims & Bakker, 2010).

In contrast, the model by Zhang and Parker (2018), following Wrzesniewski and Dutton (2001), includes cognitive crafting since “it involves altering how one frames or views their tasks or job, which is self-initiated , self-targeted, intentional, and represents meaningful changes to the job aspects ”(Zhang and Parker 2018, p.5).

In this sense, it is important to differentiate between the two forms of crafting (behavioral and cognitive), not as indicators of the same latent construct, but as aggregates.

The results of previous studies [9–11] reveal different relationship dynamics for task and relational job crafting, on the one hand, and cognitive job crafting, on the other. Additionally, recent studies also suggest that behavioral crafting (task and relational) can cause changes at the cognitive level (Unsworth, Mason, & Jones, 2004; Zhang & Parker, 2018).

b. Could cognitive crafting precede the more behavioral aspects of crafting?

Yes, it could. As we indicated in Limitations and future research section, we collected data on two waves. It would be interesting to have a third wave to confirm the mediation results strongly and to test new hypothesis. In this sense, as a dynamic process, we understand that there can be a reciprocal effect between variables, so the behavioral job crafting would affect the cognitive job crafting and vice versa. Having a third wave would allow to test this hypothesis. Nevertheless, as we have explained, the response rate in T2 was 33.3%, decreasing in almost 45%, and the participant organizations declined to collect a third wave.

c. Why wouldn’t all three aspects of job crafting be correlated predictors that are theorized to occur simultaneously? 

From the Wrzesniewski and Dutton (2001) perspective, this is the original relationships model. The novelty of our research is theoretically based on Unsworth, Mason, & Jones (2004) and Zhang & Parker (2018) revisions.

Various studies have linked job crafting with performance, including the quality of service offered by employees (Lichtehthaler and Fischbach, 2019; Rofcanin, Bakker, Berber, Gölgeci and las Heras, 2018; Rudolph et al., 2017; Tims et al., 2015; Yepes-Baldó et al., 2018). Compared to previous studies, the main novelty of our proposal lies in the fact that we analyze job crafting as a dynamic process and differentiate between behavioral job crafting, as an antecedent, and cognitive job crafting as a consequence, and as a mediator between it and the quality of care .The results of previous studies [9–11] reveal different relationship dynamics for task and relational job crafting, on the one hand, and cognitive job crafting, on the other. 

Additionally, recent studies also suggest that behavioral crafting (task and relational) can cause changes at the cognitive level (Unsworth, Mason, & Jones, 2004; Zhang & Parker, 2018).

d. Why residential care staff? I realize they are an important population and the statistics presented at the beginning of the manuscript illustrate that point, but my question was more around why job crafting would be a particularly helpful strategy with this population? Can you link the two in some way? I feel this would be instrumental in helping to build a more persuasive case for the contribution of the study 

The last few years, the healthcare sector has been characterized by reducing and slow growth of costs, but healthcare professionals need an adequate amount of resources to reach their work-related goals and offer high-quality care (French, Ikenwilo, & Scott, 2007). In this context, several authors, as Gordon, Demerouti, Le Blanc, & Bipp (2015), recognize the need for healthcare professionals to take a proactive role in shaping their future jobs to improve healthcare systems, and “job crafting is particularly interesting for health care organizations because, on one hand it can be to assure improvements on the quality of care they provide [4], and on other hand, this proactive role “can be learned and effectively transferred from training to organizational practice” (Bakker 2018, 321).

2. Regarding the method, please add the inclusion and exclusion critieria for the participants. Also, how do the characteristics of this sample match the population of residential care staff in either Spain or Sweden? To they approximate these groups? 

All the employees who were effectively working (not on leave) and who did so voluntarily participate at T1 and T2. For the final database, only the participants who participated at both T1 and T2 were included.

Although the sample obtained cannot be considered as representative due to the non-probability selection system, the demographic data for this sample is similar to that available in Spain (ACRA) and Sweden (FORTE) in terms of gender (mostly women), age, and type of work.

3. There were some aspects of the measures that were not well described. For example, I was confused by this sentence: "The internal consistency for the different Spanish versions of this scale was .85 [32] and .88 [31], and .83 for the Swedish version [32]." Is that in the present study, or in the original publications? What about the validity of the measures? 

There are different studies that use the scales in their Spanish and Swedish versions. Validity about the different scales can be seen in these studies. The studies to which we refer have been clarified in the text.

While redacting the present paper a new Spanish version was published. The Spanish versions of the instrument used in the present study [4] and in the new one are very similar. Only small changes on translation have been observed. The internal consistency for the Spanish version of this scale was .85 and .88, and .83 for the Swedish version. We considered important to mention the new published version of the instrument and its similarities with the version we used in our study.

4. For the analyses, it was odd that the models were tested as separated mediation models using PROCESS. I think the data would be more parsimoniously handled with SEM models, where you could a) include all variables simultaneously, and b) account for attenuation caused by measurement error when you model with latent variables, and c) test competing positions of the variables so that you could better establish which one is better position as a mediator, predictor, etc.

Hayes (2013) showed the similarity in results between PROCESS and an SEM program. Additionally, PROCESS estimates each equation included in a model to be tested separately (Hayes, Montoya, & Rockwood, 2017). 

Secondly, as pointed out by Hayes et al. (2017), SEM is more suitable with large samples, as it relays on large sample asymptotic theory. For this reason, as we have a sample of 226 participants, we decided to use PROCESS.

Related to latent variables, we reduced them to observed variable proxies (averages of indicators), which make them, by definition, observed and not latent (Hayes et al, 2017). 

In relation to measurement error, any analysis that can be expressed in the form of a linear regression model is not exempt of it, including SEM. On the other hand, latent variable mediation analysis may be more accurate in the estimation of effects than observed variable analysis, but less powerful in detecting them (Hayes et al, 2017).

A limitation of PROCESS is that only one exogenous variable can be entered in a single analysis. That is the reason why we used 2 separate mediation models. Additionally, as we aforementioned, previous research has arisen that the 3 components of job crafting have different relationship dynamics for task and relational job crafting, on the one hand, and cognitive job crafting, on the other. So, our decision is based on theoretical and empirical reasons.

5. On that point above, I would like to see these models compared against a model in which all three job crafting dimensions were treated as the predictor of quality of care. If this mediation model was better, then it at least provides some empirical support behind the authors’ reasoning/hypotheses, but currently there is not much. I recommend SEM as the approach to do this.

As we have different hypothesis to test, we selected PROCESS as analytical strategy to analyses the data on a more parsimonious way. The reviewer’s proposal is related to measures of fit of the model to the data, whereas PROCESS do not offer this kind of measures. However, the model built into PROCESS is saturated, so fit by some measures would be perfect when this model is estimated using SEM. Furthermore, as stated by MacCallum et al. (1993) any good fitting model typically has several minor variations with different interpretations that fit equally well (the equivalent models’ problem).

6. There is a bit too much describing of the results in the discussion. I was more interested in how this all fits with the existing literature, and a deeper discussion of the practical and theoretical implications. Especially the practical limitations feel very light, with very few details provided. I suggest you expand both of these sections considerably. 

Relation with existing literature

Research has shown that individuals who have a strong need for relatedness tend to have collectivist tendencies (Baumeister & Leary, 1995) and to help group members (Den Hartog, De Hoogh, & Keegan, 2007). Leana et al., (2009) consider that job crafting can also be a collaborative activity, as it “involves joint effort among employees in the service of changing work process” (2009, p1173). It “is not the work of an individual agent, as described by Wrzesniewski and Dutton, but instead is the work of a dyad or group of employees who together make physical and cognitive changes in the task or relational boundaries of their work" (Leana et al, 2009, 1173).

Practical implications

Our research assesses the effects of a positive organizational intervention. Specifically, our findings indicate that organizations may foster quality of care facilitating job crafting. In this sense, a practical implication of our research is that promoting job crafting in the healthcare sector might be worthwhile. The possibility of allowing workers to job craft should be highly considered by managers and organizations within the healthcare sector, especially if economic contractions are experienced, and taking into account that the healthcare professionals are working in a demanding environment. 

Additionally, following Bindl, Unsworth, Gibson, & Stride (2019), it is possible to point out the implications for job design, because “recognizing that the strength of individual needs varies across employees and allowing them the opportunity to adjust tasks, relationships, and skills in ways that enable need-fulfillment at work is important” (2019, 42-43).

Managers must acknowledge and understand employees’ perspectives, encourage self-initiative and minimize control wherever possible. In this sense, our research adds empirical evidence at the self-determination theory (Deci, Connell & Ryan, 1989), suggesting how job crafting can facilitate self-satisfaction of employees’ needs. Managers should be aware about their employees’ needs and try to encompass employees’ needs into those behaviors that are most desirable for the organization (Gagné 2003). In order to achieve this objective, it is important to assure an organizational culture oriented towards employees [35,36].

Limitation 

Our research did not consider the quality and sustainability of the job crafting developed by employees, as we have focused on employees’ job crafting and quality of care perception.

 

Reviewer #2: Dear Authors,

Thank you for submitting your manuscript and for the opportunity to review it. I certainly think it has merit but there are a number of recommendations I believe you should consider in order to improve it.

Thank you for your comments!

1. The Introduction is somewhat descriptive and could be more critical in reviewing the existing literature. First and foremost, there needs to a clear explanation into what Job Crafting is and why it is important. In particular, the manuscript alludes to a difference between the JDR perspective of crafting and that of Wrzesniewski and Dutton which is used here – however unless a reader is familiar with the job crafting literature many will not know the difference. There isn’t a clear explanation on why job crafting is important. 

Job crafting has been defined by Wrzesniewski and Dutton (2001) as the “physical and cognitive changes individuals make in the task or relational boundaries of their work” (p 179). 

Following Berg, Grant and Johnson (2010) this proposal is the understanding that employees are often interested in customizing their jobs to fit optimally their motivations, competences and desires. Organization’s management should not unilaterally decide how its employees spend their time and energy. Rather, the employees themselves should be allowed to decide what to do, creating and searching a comfortable and enjoyable context, over and beyond the job descriptions provided by the management, particularly in complex and uncertain situations (Wrzesniewski & Dutton, 2001; Ghitulescu, 2006; Leana, Appelbaum, & Shevchuk, 2009; Berg, Grant & Johnson., 2010). In this sense it is possible to consider that the job crafting emerges as a strategy for the leverage of work meaning and identity. 

2. Similarly, how does this fit in with existing theoretical models. Much of the Introduction describes existing relationships (particularly from Line 102 onwards on Page 5), but why do these relationships happen? What is the process behind it? This is particularly the case between crafting and performance (Tims et al., 2015). 

Our main objective is to test a possible mechanism to explain the relationship between job crafting and performance (in this case, quality of care) using cognitive job crafting as a mediator. 

Various studies have linked job crafting with performance, including the quality of service offered by employees (Lichtehthaler and Fischbach, 2019; Rofcanin, Bakker, Berber, Gölgeci and las Heras, 2018; Rudolph et al., 2017; Tims et al., 2015; Yepes-Baldó et al., 2018). Compared to previous studies, the main novelty of our proposal lies in the fact that we analyze job crafting as a dynamic process and differentiate between behavioral job crafting, as an antecedent, and cognitive job crafting as a consequence, and as a mediator between it and the quality of care .The results of previous studies [9–11] reveal different relationship dynamics for task and relational job crafting, on the one hand, and cognitive job crafting, on the other. 

Additionally, recent studies also suggest that behavioral crafting (task and relational) can cause changes at the cognitive level (Unsworth, Mason, & Jones, 2004; Zhang & Parker, 2018).

2.1. It would also be useful to reflect on how quality of care is different/ same to existing performance measures, and how this links in with the wider psychosocial working environment (see Teoh et al., 2019). 

During the past decades, performance indicators have become increasingly sophisticated (Majeed, Lester, Bindman, 2007), and the quality of care has been included as an indicator of performance (Westerberg and Tafvelin, 2014). Nevertheless, Marshall, Roland Brook, McGlynn and Shekelle (2003) consider that it is difficult to compare quality of care and to transfer performance indicators directly between different health systems and cultures, because each country has established different indicators. However, the quality of care is used as an indicator of patients’ outcome in studies focused in hospital care (Aiken, Clarke & Slone, 2002) and in the geriatric care (Lapointe McKenzie, Blandford, Menec, Boltz & Capezuti, 2011). 

Considering the complexity of the healthcare sector Teoh, Hassard & Cox (2019) developed a systematic review on the relationship between the working conditions and the quality of patient care. The results show, on the one hand, several investigations where an improved work environment exerted a positive effect on nurses assessed quality of care. For instance, promotion prospects, perceived salary and job security (Loerbrokset, Weigl, & Angerer, 2016. Weigl, Schneider, Hoffmann, & Angerer, 2015, Purdy, Spence Laschinger, Finegan, Kerr & Olivera, 2010) were positively related with quality of care. Nevertheless, these authors concluded that the relationships between quality of care and work environment reported divergent findings, showing the complexity of these relationships (Hannan et al. 2001, Hasson & Arnetz 2009). They recommended, among other aspects, to use multilevel or longitudinal designs, as well as mediating and moderating variables, in order to present a more realistic interpretation of these relationships. 

Quality of care could be analyzed from the nurses’ perception as an indicator of patient outcome [27,28]. The present study uses this perspective to analyze this variable 

3. Could you please expand and clarify the sentence “It is considered that cognitive crafting is a mere passive adaptation to work, so it cannot be conceived as crafting since it is not a proactive behavior of change” please? I’m not familiar with this perspective. From what I understand it appears that Zhang and Parker are saying that it isn’t job crafting, but then in the next sentence argues that is still is? 

The differentiation between behavioral and cognitive crafting arises from the existing debate between what should or should not be considered job crafting. Precisely, from the perspective of demands and resources, cognitive crafting is considered to be a mere passive adaptation to work, so it cannot be conceived as crafting since it is not a proactive behavior of change (Bakker, Tims, & Derks, 2012; Tims & Bakker, 2010).

In contrast, the model by Zhang and Parker (2018), following Wrzesniewski and Dutton (2001), includes cognitive crafting since “it involves altering how one frames or views their tasks or job, which is self-initiated , self-targeted, intentional, and represents meaningful changes to the job aspects ”(Zhang and Parker, 2018, p.5).

In this sense, it is important to differentiate between the two forms of crafting (behavioral and cognitive), not as indicators of the same latent construct, but as aggregates.

4. Page 6, 113. What are the recommendations from Westerberg and Tafvelin and how is it relevant to the point being made? When I first read this paragraph, I understood it that the intention was to use a single item measure of quality of care, but this is not the case as the study uses a 5-item measure instead. 

Even though it is not unusual to assess quality of care with a single-item [29], we follow Westerberg and Tafvelin [30] recommendations, adding information about satisfaction “with the way in which the clients were treated, kept informed and their wishes respected” [30 p464]. That is the reason why we use a scale with 5 items.

5. Page 6, line 115. It would be useful to consider reiterating these reasons, or then making them clearer earlier on, as it is not really evident. 

We hope that the changes made in the Theoretical framework and hypotheses section could clarify this point. 

6. It’s great, a real strength, to be able to carry out a longitudinal study. Nevertheless, there needs to be a rationale and description of this in the Introduction. In particularly, within the hypotheses need to make clear whether the measures are at T1 or T2 because as they currently are this appears to be a cross-sectional study. 

We agree. We changed our hypothesis to include time of each measure. Additionally, we justified our hypothesis theoretically

7. Within the material section, it is not clear if the surveys administered were done so in Swedish or in English. For the JCQ in Spanish, I don’t understand the two different Spanish versions. There is a Spanish adaption [31] that wasn’t used which I presume because it came out after the study began? Bu then the version that was used in this study, was also used in a previous study [4], so why is there a need to mention the new version develop in [31]? 

Questionnaires were administered in Spanish and Swedish. While redacting the present paper a new Spanish version was published. The Spanish versions of the instrument used in the present study and in the new one are very similar. Only small changes on translation have been observed. The internal consistency for the Spanish versions of this scale was .85 and .88, and .83 for the Swedish version. We consider it is important to mention the new published version of the instrument and its similarities with the version we used in our study.

8. It isn’t clear if the Cronbach Alpha’s in the material section refers to that from previous studies or from the current study. I understand it as the former, but I am not sure. 

We have clarified this in the text, including both, the former and the present study alpha’s

9. Page 9, Line 197 – what does medium to high mean? Are there established thresholds these are compared against? 

As we used a 5-point Likert scale, we consider 3 or higher a medium to high score. We indicated this in the text.

10. I believe that the data from Sweden and Spain were mixed-together which is fine, but I think it would be important to control for this within the analysis. 

It is correct, Sweden and Spain data were mixed-together. Unfortunately, the small number of cases in Sweden made it impossible to control for this. We include this in limitations section

10.1. Considering the longitudinal design, the relationships between cognitive crafting and quality of care still are cross-sectional. At the least, cognitive crafting at T1 should be tested as a predictor of quality of care at T2. Building on the Introduction, there needs to be some rationale as to why cognitive at crafting at T2 was used as the mediator (opposed to at T1). 

To analyze the mediation effect of CJC it is more accurate to use T2, as the antecedent (behavioral JC) is measured in T1. Nevertheless, it would be interesting to have a third wave to confirm the mediation results strongly and to test new hypothesis. In this sense, as a dynamic process, we understand that there can be a reciprocal effect between variables, so the behavioral job crafting would affect the cognitive job crafting and vice versa. Having a third wave would allow to test this hypothesis. Unfortunately, as we have explained previously, the response rate in T2 was 33.3%, decreasing in almost 45%, and the participant organizations declined to collect a third wave.

11. Considering that job crafting is strongly influenced by psychosocial and individual factors (Rudolph et al., 2016), I am left to wonder if these could/should have been included within this study. Either as possible predictors or then as control variables. At the very least, they should be acknowledged within the manuscript and discussed. 

Certainly, several author have pointed out as antecedents of job crafting proactive personality (Parker, Bindl, & Strauss, 2010; Crant, 2000, Tims, & Bakker, 2010; Grant & Ashford, 2008; Vermooten, Boonzaier, & Kidd, , 2019), knowledge, skills, and abilities (Plomp,Tims,Akkermans, Khapova, Jansen, & Bakker, 2016), self-efficacy (Crant, 2000), social support (Crant, 2000), and situational features of accountability, ambiguity, and autonomy (Grant and Ashford, 2008). Future research should include these variables, either as possible predictors or as control variables.

12. I hope you find the comments above constructive and I wish you all the best.

Your comments are really interesting, and they have allowed us to improve our paper. We hope our changes make our paper suitable for publication in PlosOne.

---

## [Decision Letter · Decision Letter 1]

18 Sep 2020

PONE-D-20-01554R1

Cognitive job crafting as mediator between behavioral job crafting and quality of care in residential homes for the elderly

PLOS ONE

Dear Dr. Yepes-Baldó,

Thank you for submitting your manuscript to PLOS ONE. After careful consideration, we feel that it has merit but does not fully meet PLOS ONE’s publication criteria as it currently stands. Therefore, we invite you to submit a revised version of the manuscript that addresses the points raised during the review process.

The manuscript has improved, but the authors need to address the issues raised by both reviewers. In addition to the comments of the reviewers (at the bottom of this letter), I would like the authors to address the following points concerning their work:

1.     The authors have not yet adequately explained why  cognitive crafting would act as a mediator between the more behavioral aspects of crafting and quality of care.  Could cognitive crafting precede the more behavioral aspects of crafting?

2.    The authors have not yet answered the question of a previous reviewer.  For the analyses, why were the models tested as separated mediation models using PROCESS? As noted by a previous reviewer,  the data would be more parsimoniously handled with SEM models, where they could a) include all variables simultaneously, and b) account for attenuation caused by measurement error when you model with latent variables, and c) test competing positions of the variables so that you could better establish which one is better position as a mediator, predictor, etc. So, the authors should address this issue. 

3.     What are the practical implications for the people involved in their research? How should they change the way they work based on the results of the research? We need to know the implications of the research for work practices. 

We look forward to receiving your revised manuscript.

Kind regards,

Anthony Montgomery

Academic Editor

PLOS ONE

Reviewers' comments:

Reviewer's Responses to Questions

**Comments to the Author**

1. If the authors have adequately addressed your comments raised in a previous round of review and you feel that this manuscript is now acceptable for publication, you may indicate that here to bypass the “Comments to the Author” section, enter your conflict of interest statement in the “Confidential to Editor” section, and submit your "Accept" recommendation.

Reviewer #2: (No Response)

Reviewer #3: All comments have been addressed

2. Is the manuscript technically sound, and do the data support the conclusions?

Reviewer #2: Yes

Reviewer #3: Yes

3. Has the statistical analysis been performed appropriately and rigorously? 

Reviewer #2: Yes

Reviewer #3: Yes

4. Have the authors made all data underlying the findings in their manuscript fully available?

Reviewer #2: Yes

Reviewer #3: Yes

5. Is the manuscript presented in an intelligible fashion and written in standard English?

Reviewer #2: No

Reviewer #3: Yes

6. Review Comments to the Author

Reviewer #2: Thank you for your revisisions and addressing my previous comments. These are mostly addressed and I have a few more clarification points:

The inclusion of the Wrzesniewski and Dutton definition should be integrated into one of the paragraphs and not be a standalone sentence.

I don’t feel the second original point on why do these relationships happen (from line 80 onwards) is evident. The revisions are still descriptive, although the work of Ashforth and Kreiner helps with this. But this could still be unpacked further, and especially in explaining how and why job crafting is associated with quality of care.

In terms of the Teoh et al. paper, the intention was not so much to summarise the findings of it, as actually the description and links between psychosocial working conditions and quality of care is not directly relevant. Instead, that paper highlights how quality of care can be operationsed in different ways and that it rather complex. What is needed here is for the paper to consider what is quality of care and how it is measured here relates to performance more generally and/or other measures of quality of care. The revised new paragraph (Line 131, During the past decades…) does a decent job at this, but it needs to better link in with the next paragraph on the review. This paper does highlight the review’s call for longitudinal and mediating research, which this study actually answers, and it would be good to use this as additional justification to strengthen the need for this study.

With regards to the point “As we used a 5-point Likert scale, we consider 3 or higher a medium to high score. We indicated this in the text”, I don’t think this is needed. What is the justification of the 3 and above as a medium to high score? Is this a cut-off determined as part of the questionnaire or something that was decided here? If there is no set comparison (normed) score than its not appropriate to say its high.

There are elements of duplication that can be removed from this paper. For example there is no need to describe the five-point QoC scale in the Introduction and then again in the Method section. Equally if Cronbach Alpha scores are available in the Table 2 and it is mentioned that they all have good reliability then there is no need to mention them also in the measures section.

For H1 and H2, was it hierarchical multiple regressions that were carried out or simple linear regressions? If it is the former, which it should be, then this should be made clearer and the control variables also acknowledged.

Cognitive job crafting T2 should be made clearer within the hypotheses. While ideally there would have been three time points (but two is really good as well – especially as it’s a year apart), as this is not the case here there should be some consideration as to why cognitive job crafting at Time 2 was used and not that from Time 1.

Page 4 line 66: “the healthcare sector has been characterized by reducing and slow growth of costs” – this sentence isn’t clear, is it reducing or is it having slow growth?

Reviewer #3: I consider this to be a very interesting job as it applies job crafting to a group of professionals who were traditionally analyzed from the wrong perspectives.

The design of the work is very appropriate as it combines the cross-cultural perspective with an exhaustive control over the variables of the study.

I have a question regarding the tests they have used:

a. In which languages have the work questionnaires been applied? Spanish and Swedish? (page 10)

b. Did they apply the questionnaires in "paper and pencil" format? (page 9)

c. With an "online" application of the questionnaires could they have achieved greater success in answering? (page 11)

7. PLOS authors have the option to publish the peer review history of their article (what does this mean?). If published, this will include your full peer review and any attached files.

Reviewer #2: No

Reviewer #3: No

---

## [Author Response · Author response to Decision Letter 1]

2 Oct 2020

ANSWERS TO EDITOR

Following the instructions, we attached the answers to reviewers file to our submission, including our answer to the editor comments. Nevertheless, you can see here our answers to editor comments again.

COMMENT 1. The authors have not yet adequately explained why cognitive crafting would act as a mediator between the more behavioral aspects of crafting and quality of care. Could cognitive crafting precede the more behavioral aspects of crafting?

AUTHORS ANSWER: Recent studies suggest that behavioral crafting (task and relational) can cause changes at the cognitive level (Unsworth, Mason, & Jones, 2004; Zhang & Parker, 2018). In this sense, Unsworth et al. (2004) pointed out that “group members cognitively frame their comparisons such that they compare themselves with other groups on dimensions that represent in-group strengths (…). For example, members of a highly productive group would tend to evaluate groups in terms of their productivity, but a less productive group might choose to focus on other attributes, such as friendliness” (p.5). In our research, following this suggestion, we consider that employees could cognitive reframe the meaning of their jobs by means of the tasks (Task crafting) they develop and the relationships they have at work (Relational crafting) (Theoretical framework and hypotheses section)

Various studies have linked job crafting with performance, including the quality of service offered by employees (Lichtehthaler and Fischbach, 2019; Rofcanin, Bakker, Berber, Gölgeci and las Heras, 2018; Rudolph et al., 2017; Tims et al., 2015; Yepes-Baldó et al., 2018). Compared to previous studies, the main novelty of our proposal lies in the fact that we analyze job crafting as a dynamic process and differentiate between behavioral job crafting (tasks and relational crafting), as an antecedent, and cognitive job crafting as a consequence, and as a mediator between it and the quality of care .The results of previous studies [9–11] reveal different relationship dynamics for task and relational job crafting, on the one hand, and cognitive job crafting, on the other. 

COMMENT 2. The authors have not yet answered the question of a previous reviewer. For the analyses, why were the models tested as separated mediation models using PROCESS? As noted by a previous reviewer, the data would be more parsimoniously handled with SEM models, where they could a) include all variables simultaneously, and b) account for attenuation caused by measurement error when you model with latent variables, and c) test competing positions of the variables so that you could better establish which one is better position as a mediator, predictor, etc. So, the authors should address this issue. 

AUTHORS ANSWER: Hayes (2013) showed the similarity in results between PROCESS and an SEM program. Additionally, PROCESS estimates each equation included in a model to be tested separately (Hayes, Montoya, & Rockwood, 2017). 

Secondly, as pointed out by Hayes et al. (2017), SEM is more suitable with large samples, as it relays on large sample asymptotic theory. For this reason, as we have a sample of 226 participants, we decided to use PROCESS.

Related to latent variables, we reduced them to observed variable proxies (averages of indicators), which make them, by definition, observed and not latent (Hayes et al, 2017). 

In relation to measurement error, any analysis that can be expressed in the form of a linear regression model is not exempt of it, including SEM. On the other hand, latent variable mediation analysis may be more accurate in the estimation of effects than observed variable analysis, but less powerful in detecting them (Hayes et al, 2017).

A limitation of PROCESS is that only one exogenous variable can be entered in a single analysis. That is the reason why we used 2 separate mediation models. Additionally, as we aforementioned, previous research has arisen that the 3 components of job crafting have different relationship dynamics for task and relational job crafting, on the one hand, and cognitive job crafting, on the other. So, our decision is based on theoretical and empirical reasons.

COMMENT 3. What are the practical implications for the people involved in their research? How should they change the way they work based on the results of the research? We need to know the implications of the research for work practices. 

AUTHORS ANSWER: Following a psychosocial perspective, we considered important to point out the context key aspects for human resource management. In order to allow people “to change the way their work” we focused on the organizational context and managerial practices. We consider that it is important to guarantee the working conditions that allow employees turn the job they have into the job they want (Wrzesniewski , Berg, and Dutton, 2010).

We added this consideration in the text (Implications for theory and research).

---

## [Decision Letter · Decision Letter 2]

30 Nov 2020

Cognitive job crafting as mediator between behavioral job crafting and quality of care in residential homes for the elderly

PONE-D-20-01554R2

Dear Dr. Yepes-Baldó,

We’re pleased to inform you that your manuscript has been judged scientifically suitable for publication and will be formally accepted for publication once it meets all outstanding technical requirements.

Kind regards,

Ting Ren

Academic Editor

PLOS ONE

Additional Editor Comments (optional):

Reviewers' comments:

Reviewer's Responses to Questions

**Comments to the Author**

1. If the authors have adequately addressed your comments raised in a previous round of review and you feel that this manuscript is now acceptable for publication, you may indicate that here to bypass the “Comments to the Author” section, enter your conflict of interest statement in the “Confidential to Editor” section, and submit your "Accept" recommendation.

Reviewer #2: (No Response)

Reviewer #3: All comments have been addressed

2. Is the manuscript technically sound, and do the data support the conclusions?

Reviewer #2: Yes

Reviewer #3: (No Response)

3. Has the statistical analysis been performed appropriately and rigorously? 

Reviewer #2: Yes

Reviewer #3: Yes

4. Have the authors made all data underlying the findings in their manuscript fully available?

Reviewer #2: Yes

Reviewer #3: Yes

5. Is the manuscript presented in an intelligible fashion and written in standard English?

Reviewer #2: Yes

Reviewer #3: Yes

6. Review Comments to the Author

Reviewer #2: Dear Authors,

Thank you for addressing the comments from the last revision. I only have one minor point. The included sentence in line 120 with the quote by Unsworth et al. could perhaps be paraphrased and not kept as a direct quotation. Also, both sentence start of with point/pointed out.

Returning to the point on PROCESS mediation and SEM, your explanation is fine but I still think SEM has its advantages and not doing so remains a limitation that should be acknowledged.

Reviewer #3: The work responds to the quality that is demanded in the magazine. The authors' answers to my questions are correct and adequate

7. PLOS authors have the option to publish the peer review history of their article (what does this mean?). If published, this will include your full peer review and any attached files.

Reviewer #2: No

Reviewer #3: No

---

## [Editor Report · Acceptance letter]

3 Dec 2020

PONE-D-20-01554R2 

Cognitive job crafting as mediator between behavioral job crafting and quality of care in residential homes for the elderly 

Dear Dr. Yepes-Baldó:

I'm pleased to inform you that your manuscript has been deemed suitable for publication in PLOS ONE. Congratulations! Your manuscript is now with our production department. 

Kind regards, 

on behalf of

Dr. Ting Ren 

Academic Editor

PLOS ONE